# Open structure and gating of the *Arabidopsis* mechanosensitive ion channel MSL10

Jingying Zhang[1,2,3], Grigory Maksaev [1,2] & Peng Yuan [1,2,3,4] ✉

Plants are challenged by drastically different osmotic environments during growth and development. Adaptation to these environments often involves mechanosensitive ion channels that can detect and respond to mechanical force. In the model plant *Arabidopsis thaliana*, the mechanosensitive channel MSL10 plays a crucial role in hypo-osmotic shock adaptation and programmed cell death induction, but the molecular basis of channel function remains poorly understood. Here, we report a structural and electrophysiological analysis of MSL10. The cryo-electron microscopy structures reveal a distinct heptameric channel assembly. Structures of the wild-type channel in detergent and lipid environments, and in the absence of membrane tension, capture an open conformation. Furthermore, structural analysis of a non-conductive mutant channel demonstrates that reorientation of phenylalanine side chains alone, without main chain rearrangements, may generate the hydrophobic gate. Together, these results reveal a distinct gating mechanism and advance our understanding of mechanotransduction.

During growth and development, plants constantly experience a wide variety of exogenous and endogenous mechanical stress such as touch, gravity, wind, flood, wound, pathogen invasion, and osmotic pressure[1]. One way to perceive and respond to these mechanical cues is to engage mechanosensitive (MS) ion channels to convert force into electrical and chemical signals. Carnivorous plants, such as the Venus flytrap (*Dionaea muscipula*), involve MS channels in the sensory hairs on the leaf blade to sense the touch of prey and elicit action potentials, eventually leading to rapid leaf closure[2]. An array of MS channels, including two-pore potassium[3], OSCA[4–7], Piezo[8], MCA[9,10], and MSL channels[1,11], have been identified in plants to perform specialized physiological functions. Remarkably, in the model plant *Arabidopsis thaliana*, ten distinct MSL channels (MSL1-10), which are homologous to the prokaryotic mechanosensitive channel of small conductance (MscS)[12–14], are expressed in different tissues and cellular compartments[1,11]. For instance, MSL1, MSL2, and MSL8 have been found to localize to the mitochondria[15], chloroplast[16], and pollen membranes[17], respectively. MSL1 has been proposed to regulate the membrane potential and control redox homeostasis under abiotic stress in mitochondria[15], whereas MSL8 critically manages osmotic stress in pollen hydration and germination[17].

Amongst the *Arabidopsis* MSL channels, MSL10 is enigmatic in that it has two separable functional modalities, regulated cell death signaling conferred by its N-terminal soluble 'death' domain and force-activated ion channel activity imparted by its C-terminal portion[18]. When heterologously expressed in *Xenopus* oocytes, MSL10 is indeed activated by membrane stretch and displays a preference for anions[19]. In *Arabidopsis*, MSL10 is primarily expressed in the shoot apex, root tips, and vascular tissues[20,21], playing a critical role in adaptation to hypo-osmotic shock and induction of programmed cell death[18,22]. Moreover, recent studies have suggested that MSL10 participates in wound signaling and perception of mechanical oscillations caused by the wind[23,24]. On the protein sequence level, MSL10 is closely related to MSL9 (75% sequence identity) and forms heteromeric channels with MSL9 in the wild-type plant[20]. In addition, MSL10 is homologous to Flycatcher1 (*Dm*MSL10/FLYC1, 48% sequence identity), a recently identified MS ion channel involved in prey recognition in the Venus flytrap[25–27].

[1]Department of Cell Biology and Physiology, Washington University School of Medicine, Saint Louis, MO, USA. [2]Center for the Investigation of Membrane Excitability Diseases, Washington University School of Medicine, Saint Louis, MO, USA. [3]Department of Pharmacological Sciences, Icahn School of Medicine at Mount Sinai, New York, NY, USA. [4]Department of Neuroscience, Icahn School of Medicine at Mount Sinai, New York, NY, USA. ✉e-mail: peng.yuan@mssm.edu

To better understand MS channels pivotal for plant physiology, we determined the cryo-electron microscopy (cryo-EM) structures of the wild-type MSL10 channel from *Arabidopsis thaliana* (*At*MSL10) as well as its functional mutants. Strikingly, an open structure of the wild-type channel solubilized in detergent micelles or embedded in lipid nanodiscs is achieved in the absence of applied force. By comparison, the structure of an inactive variant reveals how the ion conduction path can be potentially closed by side chain reorientation of a gating phenylalanine residue, without global structural rearrangements of the channel.

## Results

### Overall structure

We expressed the full-length wild-type *At*MSL10 channel, consisting of 734 amino acids, in *Pichia pastoris* and purified the channel to homogeneity for single-particle cryo-EM analysis (Supplementary Fig. 1). We obtained 3D reconstruction at a nominal resolution of ~3.7 Å, with C7 symmetry applied (Fig. 1a), built an atomic model facilitated by the AlphaFold prediction of a protomer[28], and refined the structure to good stereochemistry (Supplementary Fig. 2, Supplementary Table 1). The channel core, consisting of TM3-TM6 and the cytoplasmic C-terminal domain, was well resolved in the cryo-EM density map and allowed unambiguous placement of most of the side chains. However, the N-terminal 'death' domain (residues 1–165), part of the cytoplasmic domain between TM4 and TM5, including residues 335–386, 397–431, and 469–480, and the C-terminal end (residues 732–734), were not resolved in the reconstruction, and thus were absent in the final refined atomic model. The linker domain between TM4 and TM5 and most of the residues in the peripheral TM1–TM2 helices were modeled as poly-alanine because of weak side-chain densities.

*At*MSL10 assembles as a symmetric heptamer in that the cryo-EM reconstructions without applied symmetry (C1 symmetry) and with imposed C7 symmetry are essentially identical. The overall architecture shows similarity to other MscS homologs (Fig. 2a), including *E. coli* MscS (*Ec*MscS)[14,29], YnaI[30], MscK[31], and *A. thaliana* MSL1 (*At*MSL1)[32,33], and most closely resembles that of the Venus flytrap *Dm*MSL10/FLYC1[25]. Analogous to *Dm*MSL10/FLYC1, each subunit of the heptameric *At*MSL10 channel is composed of a transmembrane domain (TMD) with six membrane-spanning helices (TM1-TM6) and a characteristic C-terminal cytoplasmic domain (CTD) shared by all MscS homologs (Figs. 1, 2a). Other structural features common for MscS homologs, including a sharply kinked pore-lining helix (TM6a and TM6b), nearly single-file arrangement of the transmembrane

helices, and side portals in the cytoplasmic CTD critical for ion permeability and conduction[25,33,34], are also present in *At*MSL10 (Figs. 1, 2a). Notably, the arrangement of TMDs creates large cavities between adjacent subunits (Fig. 1a), which are presumably filled with lipid molecules in a biological membrane. The relaxed inter-subunit packing of TMDs seems to be an additionally shared structural feature amongst MscS homologs including MSL1, MscK, and *Dm*MSL10/FLYC1[25,31–33].

The *At*MSL10 structure also reveals a cytoplasmic linker domain (CLD) connecting TM4 and TM5, analogous to that observed in *Dm*MSL10/FLYC1 (Fig. 2)[25]. The seven CLDs loosely encompass the central cytoplasmic cage, making marginal contacts with the remaining of the channel. Densities corresponding to the peripheral CLDs are less well-resolved in the cryo-EM reconstruction compared with densities of the channel core, indicating dynamic nature of the CLDs. Interestingly, in *Dm*MSL10/FLYC1, two distinct conformations, the 'up' and 'down' states, were observed for the seven CLDs (Fig. 2), and the consensus reconstruction rendered a ratio of 6 'up' to 1 'down' in the heptameric channel[25]. In contrast, reconstructions of *At*MSL10 in C1 or C7 symmetry indicated that all CLDs adopt the 'down' configuration. To further assess whether a portion of the CLDs in the heptameric *At*MSL10 channel displays a different conformation, we applied symmetry expansion and focused classification of protomers and found no evidence of the 'up' configuration for the CLDs (Supplementary Fig. 1).

Notably, in *Dm*MSL10/FLYC1, a salt bridge between R334 and D598 plays an important role in stabilizing the 'up' configuration of the CLD (Fig. 2b)[25]. Introduction of a charge reversal mutation (R334E), which disrupts this critical interaction and thus likely favors the 'down' configuration, stabilized the open state of the channel[25]. Interestingly, the corresponding residues in *At*MSL10, N316 and D579 (Fig. 2b, Supplementary Fig. 3), are incapable of forming analogous inter-domain salt-bridge interactions, explaining the lack of an 'up' conformation of the CLDs in *At*MSL10 and highlighting unique structural features and potentially distinct gating schemes between these two structurally closely related channels.

### Open conformation of the wild-type MSL10

Consecutive bulky hydrophobic residues in the pore-lining helices, L105 and L109 in *Ec*MscS, V319 and F323 in *At*MSL1, V568 and F572 in *Dm*MSL10/FLYC1, and V921 and F925 in *Ec*MscK, respectively, are presumed to form a hydrophobic gate in the resting state and expand upon channel opening[25,31,32,35]. The corresponding residues in this region in *At*MSL10 are V549 and F553 (Fig. 3a, b). To evaluate the conformational state of the wild-type *At*MSL10 structure, we

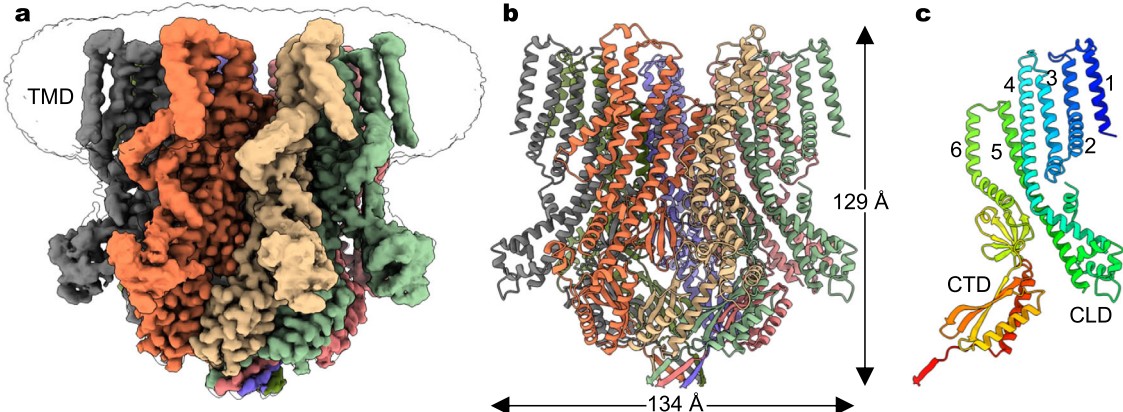

**Fig. 1 | Cryo-EM structure of *At*MSL10. a** Cryo-EM reconstruction of the wild-type full-length channel in detergents, with each subunit uniquely colored. The unsharpened map contoured at a lower level (in black) illustrates the detergent micelle densities surrounding the transmembrane domain (TMD). Notably, densities for the N-terminal 'death' domain (residues 1–165) are not resolved. **b** The heptameric channel architecture. **c** A single channel subunit, with TM helices 1–6, cytoplasmic linker domain (CLD), and C-terminal cytoplasmic domain (CTD) highlighted.

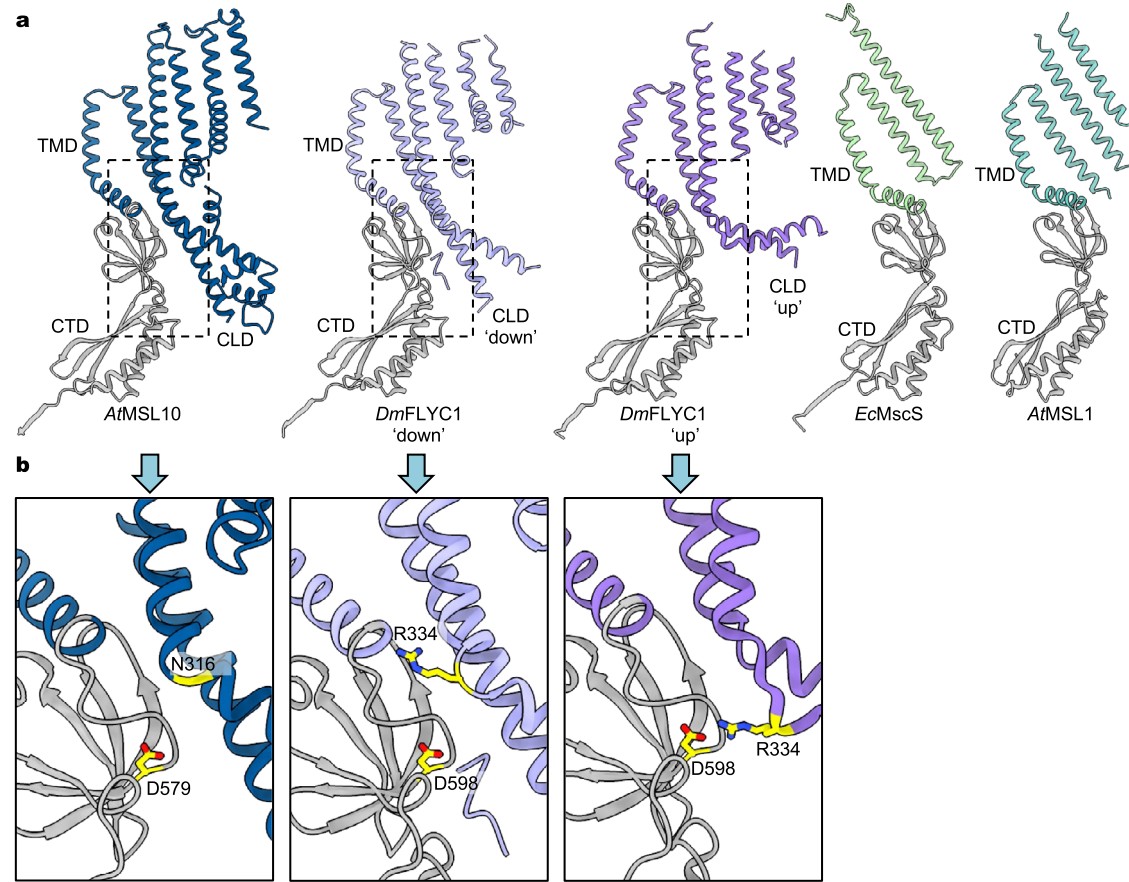

**Fig. 2 | Structural comparison of MscS homologs. a** Protomer structures of *At*MSL10, the 'down' and 'up' conformations of *Dm*MSL10/FLYC1 (PDB: 7N5D), *Ec*MscS (PDB: 6RLD), and *At*MSL1 (PDB: 6VXM). TMD, CTD, and CLD are labeled. **b** Closeup views of the boxed regions from (**a**). The corresponding residues forming a salt bridge in the 'up' conformation of *Dm*FLYC1 are shown as sticks. The side chain of N316 in *At*MSL10, which corresponds to R334 in *Dm*FLYC1, was not resolved in the cryo-EM density map. The cytoplasmic linker domain (CLD) between TM4 and TM5 in *At*MSL10 resembles the 'down' conformation of CLD in *Dm*FLYC1.

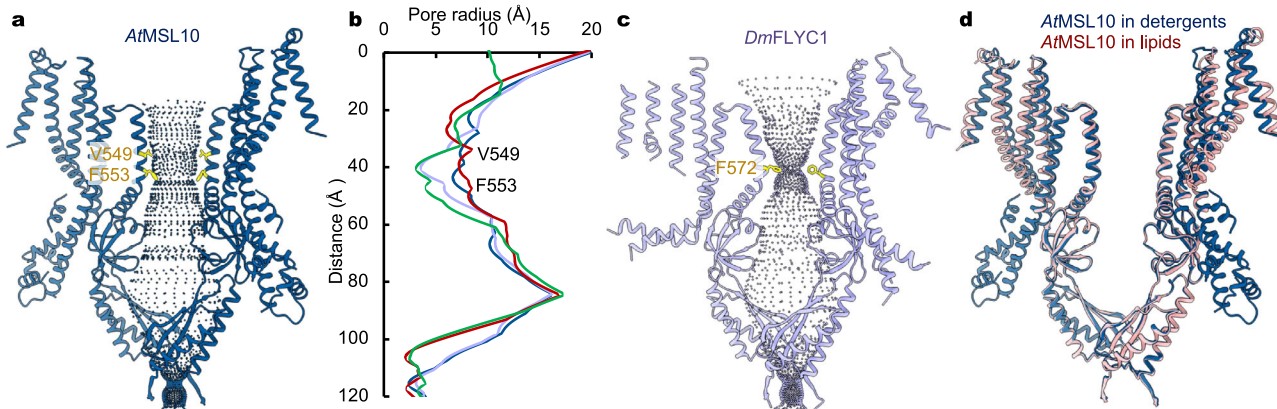

**Fig. 3 | Open conformation of the wild-type *At*MSL10. a** The ion conduction pore of *At*MSL10. Only two opposing subunits are shown for clarity, and blue dots outline the central pore. The narrow positions at V549 and F553 are highlighted. **b** Pore dimension. The pore profile of the wild-type *At*MSL10 (dark blue) is compared with those of *Dm*MSL10/FLYC1 (light blue, PDB:7N5D) and open (red, PDB:2VV5) and closed (green, PDB: 6RLD) *Ec*MscS. **c** The ion pore of *Dm*MSL10/FLYC1, with the narrow constriction at F572 labeled. **d** Superposition of the structures of *At*MSL10 in detergents (dark blue) and in saposin lipid nanoparticles (salmon).

calculated the pore radius using program HOLE[36]. Strikingly, pore radius estimation indicated a wide-open central ion conduction path (Fig. 3b). The narrowest opening of the pore, defined by F553 near the helical kink in TM6, measures ~6.6 Å in radius and is comparable to the open pore dimension of *Ec*MscS[35]. In contrast, the wild-type *Dm*MSL10/FLYC1 channel, which is the closest ortholog of *At*MSL10, has a minimum radius of 3.5 Å at the equivalent hydrophobic gate[25]. Thus, comparison of the ion-conduction pore dimensions of these channels indicates that the *At*MSL10 structure likely represents an open conformation, which is remarkable because the wild-type *At*MSL10 structure was determined in the absence of applied membrane tension that is required for channel activation.

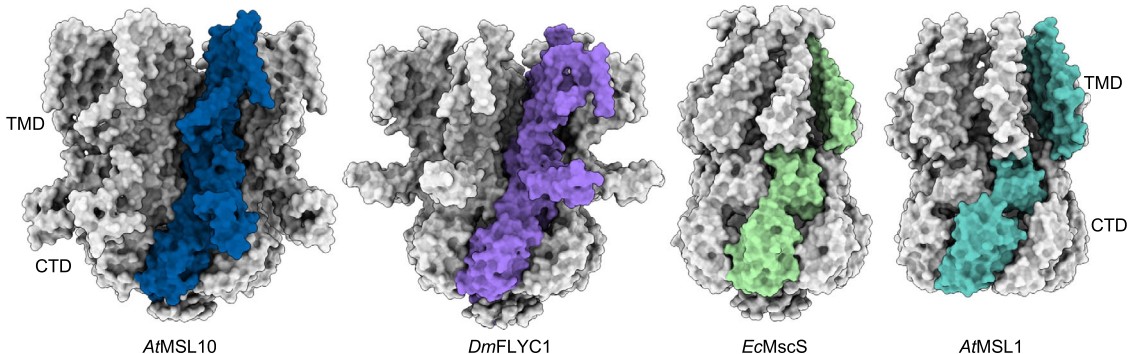

**Fig. 4 | Domain organization of MscS homologs.** The structures of *At*MSL10, *Dm*MSL10/FLYC1 (PDB: 7N5D), *Ec*MscS (PDB: 6RLD) and *At*MSL1 (PDB: 6VXM) are shown. In each of the heptameric channels, a single subunit is uniquely colored to illustrate domain arrangements.

Why does the wild-type channel adopt an open conformation, without applied mechanical force? Examination of *At*MSL10 channel activity provides a hint. When heterologously expressed in *Xenopus* oocyte, the wild-type *At*MSL10 displays asymmetric channel opening and closing in excised membrane patches[19]. The activated channels close at substantially lower membrane tensions than they open, and some channels remain open after release of applied pressure in the recording pipettes[19]. Therefore, the activated channels are inclined to stay open, which provides a plausible explanation why an open structure might be obtained in the absence of force. It is likely that the channel maintains a conductive conformation once activated by mechanical stress introduced during biochemical preparation.

The open *At*MSL10 structure was determined in detergent micelles that were essentially devoid of membrane lipids. Consistently, we did not observe lipid-like densities surrounding the channel. Notably, previous studies have demonstrated that purified wild-type *Ec*MscS in detergents adopts an open conformation when lipids are sufficiently removed and that adding lipids back to purified *Ec*MscS in detergents restores the closed conformation[37]. These observations suggest that departure of lipids bound to a MS channel, which would occur under elevated membrane tension, likely favors an open conformation. Therefore, we asked whether *At*MSL10 embedded in a lipid environment, without applied tension, renders a closed conformation. Owing to the limited dimension of lipid nanodiscs generated by membrane scaffold proteins[38], we decided to reconstitute *At*MSL10 in a lipid environment using saposin lipid nanoparticles[39] and determined the cryo-EM structure at an overall resolution of ~3.6 Å (Fig. 3d, Supplementary Fig. 4), which is similar to the structure of *At*MSL10 in detergents (root-mean-square deviation (r.m.s.d) of ~1.6 Å for all Cα atoms). The most pronounced differences come from the outer TM helices, which undergo slight rotation and outward movement from the central pore, in comparison with the structure of *At*MSL10 in detergents (Fig. 3d). Interestingly, the CLD between TM4 and TM5 was completely disordered in the lipid environment, further indicating structural flexibility of this peripheral domain. Remarkably, no discernible densities corresponding to lipids bound to the channel were observed, though the channel was fully embedded in lipids. Therefore, the structures of *At*MSL10 in detergent and lipid environments both represent an open, conductive state, further indicating that the wild-type *At*MSL10 channel, unlike other MS channels, uniquely favors an open conformation.

### Distinct domain arrangement
Despite that the *At*MSL10 protomer has a similar fold as those of *Ec*MscS, *Ec*MscK, and *At*MSL1, the heptameric *At*MSL10 channel assembly has a distinct domain arrangement (Fig. 4), which is analogous to that of the recently characterized *Dm*MSL10/FLYC1[25]. In *At*MSL10 and *Dm*MSL10/FLYC1, the peripheral TM helices (TM1–TM5)

interact with the CTD from the same protomer. In contrast, in the wild-type structures of *Ec*MscS and *At*MSL1, the TMD interacts with the CTD from an adjacent subunit in a domain-swapped configuration (Fig. 4). In addition, in *At*MSL10 and *Dm*MSL10/FLYC1, the unique TM4–TM5 linker domain contacts the cytoplasmic CTD from the same subunit, which likely contributes to the distinct non-domain swapped architecture. The distinct structural features in *At*MSL10 and *Dm*MSL10/FLYC1, including the CLD between TM4 and TM5 and the non-domain swapped arrangement between the TMD and CTD, suggest a gating scheme deviating from those proposed for *Ec*MscS, *Ec*MscK, and *At*MSL1[31,32,35,40,41].

In addition to the characteristic TM4-TM5 CLD, we observed an inter-subunit salt bridge between E534 and K539 at the extracellular end of TM5-TM6, which is unique to *At*MSL10 (Supplementary Figs. 3, 5). Thus, we asked whether the observed salt-bridge interaction contributes to unique structural and functional properties of the wild-type *At*MSL10 channel. We introduced two single-point charge reversal mutations at these two positions, E534K or K539E, to disrupt the inter-subunit salt bridge interaction. Surprisingly, both mutations virtually maintained the wild-type channel activity, as assessed in excised membrane patches from *Xenopus* oocytes (Supplementary Fig. 5). We determined the cryo-EM structure of K539E at ~3.7 Å resolution (Supplementary Fig. 6), which represents an open conformation nearly identical to that of the wild type (r.m.s.d of ~0.3 Å for all Cα atoms). Interestingly, densities corresponding to the TMD and the TM4-TM5 linker domain are better resolved in the K539E mutant than in the wild type. Thus, the extracellular E534-K539 salt bridge between two adjacent subunits in the heptameric *At*MSL10 channel is not required for the non-domain swapped arrangement or channel gating by membrane tension. Supporting this notion, *Dm*MSL10/FLYC1, which also adopts a non-domain swapped arrangement, does not contain an equivalent salt bridge in the corresponding region.

### Gating transition
Like other MscS homologs, residues in the pore-lining helix of *At*MSL10 critically influence channel gating, and perturbations alter channel activity[42]. Of particular interest, transient overexpression of the G556V mutation, near the helical kink in TM6, in tobacco epidermal cells resulted in a well-expressed protein as assessed by confocal imaging and the planta cell death assay[42]. Transient overexpression of G556V in *Xenopus oocytes* also demonstrated protein expression in the plasma membrane. However, recordings from excised membrane patches showed no detectable currents of G556V under applied tension, indicating that the mutation stabilizes a non-conducting conformation[42]. To better understand the open-to-closed transition of *At*MSL10, we determined the cryo-EM structure of G556V at a nominal resolution of ~3.5 Å (Fig. 5, Supplementary Fig. 7). The global structure of G556V is almost identical to that of the wild type (r.m.s.d of ~1.1 Å for all Cα

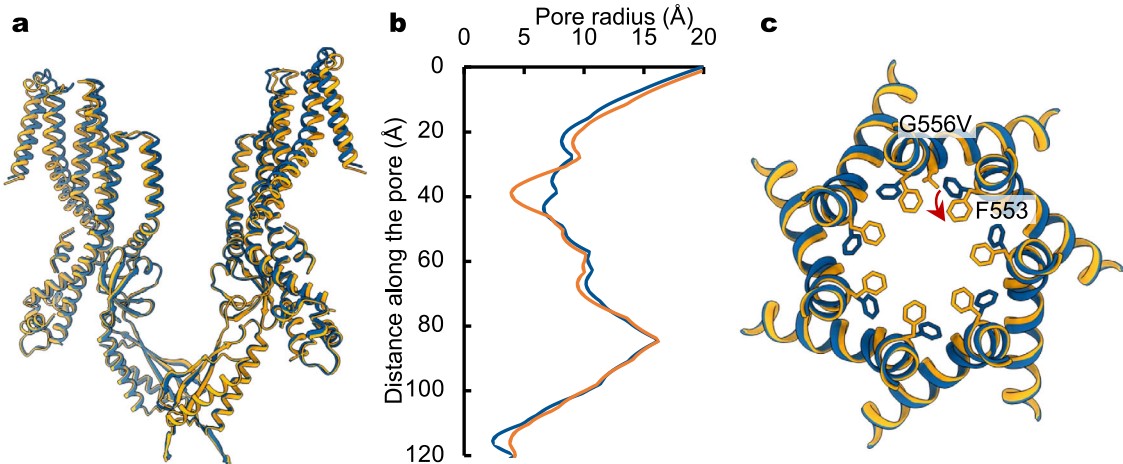

**Fig. 5 | Hydrophobic gate closure. a** Overlay of the structures of the wild-type *At*MSL10 (blue) and G556Vmutant (orange). **b** Comparison of the ion pore profiles of the wild-type (blue) and G556V mutant (orange) channels. **c** Superposition of the pore lining helices of the wild type (blue) and G556V (orange). Amino acids 553 and 556 are highlighted as sticks. The side chain reorientation of F553 is indicated by a red arrow.

atoms). Importantly, pronounced local conformational changes occur at the F553 hydrophobic gate, though the pore-lining helices align well with those of the wild type and the helical kinks are maintained (Fig. 5c). Introduction of a bulkier hydrophobic residue, valine, at the position of G556 pushes the aromatic side chain of F553 toward the central pore axis (Fig. 5c). Consequently, reorientation of the pheny-lalanine side chain reduces the pore radius from 6.6 to 3.9 Å at the position of the F553 ring (Fig. 5b), which is now remarkably similar to the dimension of the corresponding F572 ring in the wild-type *Dm*MSL10/FLYC1 structure (3.5 Å in radius)[25]. Together with the lack of activating currents of the G556V variant in excised membrane pat-ches, we conclude that this conformation represents a non-conduct-ing, closed state. Interestingly, in *Dm*MSL10/FLYC1, the side chains of the equivalent F572 residues adopt different conformations in the protomers with 'up' and 'down' configurations and experience rotamer switching in all-atom molecular dynamics (MD) simulations[25]. There-fore, reorientation of the side chains of the phenylalanine residues could represent the gating transitions between the closed and open states in both *At*MSL10 and *Dm*MSL10/FLYC1 channels.

## Discussion

Mechanotransduction mediated by MS ion channels is pivotal for all kingdoms of life. To fully understand the molecular basis of this process requires visualization of MS channels in distinct functional states along their gating cycles. However, it is inherently challenging to obtain open conformations of any MS channels because mechanical force could not be readily applied in structural biology techniques. To circumvent this challenge, previous studies have mainly relied on obtaining open structures through functional identification of gain-of-function mutants that stabilize the open states[32,35]. In this work, our studies of the *Arabidopsis* MSL10 channel have provided unique insights into its architecture and mechan-otransduction mechanism. Unprecedently, we have visualized the open conformation of the wild-type *At*MSL10 channel, representing a rare success in elucidating the open state of a wild-type MS channel.

The full-length *At*MSL10 channel was subjected to structural ana-lysis, but the N-terminal 'death' domain (NTD), consisting of amino acids 1-165, could not be resolved in the cryo-EM densities. This is consistent with the finding that the NTD is an intrinsically disordered region (IDR) that interacts with the C-terminal domain of MSL10 to regulate cell death signaling and ion channel activity[43,44]. This notion is also reminiscent of the TRPV4 ion channel that is involved in thermo- and osmoregulation, in which the N-terminal IDR modulates channel

activity via dynamic long-range interactions[45]. Together, these obser-vations highlight the critical roles of IDRs in ion channel physiology.

Amino acid sequence homology between *At*MSL10 and the pro-totypical *Ec*MscS is limited, even in the most conserved regions including the pore-lining helix and subsequent β-domain[42]. The pore-lining helix TM3 in *Ec*MscS is primarily constituted by hydrophobic residues that are enriched in small-sized amino acids such as alanine and glycine. In contrast, the pore-lining helix TM6 in *At*MSL10 contains multiple bulky hydrophobic residues as well as polar or charged resi-dues. The only notable similarities are within a short hydrophobic stretch immediately preceding the helical kink (G113 in *Ec*MscS and G556 in *At*MSL10). In *Ec*MscS, G113 is critical for inactivation[46], and the G113A mutant does not inactivate under application of sustained membrane tension[47]. The equivalent G556A mutant in *At*MSL10 essentially maintains wild-type channel activity with a slightly lower unitary conductance, and a bulkier mutation G556V abolishes channel activity[42]. Together these contrasting observations hint at a distinct gating scheme in *At*MSL10.

Structural and functional characterization of *At*MSL10, as well as comparison with *Dm*MSL10/FLYC1, suggests a unique gating mechanism primarily governed by reorientation of the aromatic side chain of a phenylalanine residue located at the hydrophobic gate. The wild-type *At*MSL10 structure has a wide-open pore, whereas a non-conducting mutant G556V renders a much narrower pore created by repositioning of the side chain of F553, which appears to constitute the hydrophobic gate in the closed state. Substitution of F553 with tryp-tophan or leucine has demonstrated that the bulkiness of the side chain negatively correlates with unitary conductance, whereas the F553V mutant abolishes channel activity[42]. Therefore, the F553 position is critically involved in gating and determining unitary conductance, which is in line with our proposition of F553 functioning as a hydro-phobic gate. In *Dm*MSL10/FLYC1, individual subunits in the asym-metric heptameric channel exhibit two distinct conformations, in which the TM4-TM5 linkers adopt the 'up' or 'down' configuration[25]. Notably, the equivalent phenylalanine F572 residues at the gate in *Dm*MSL10/FLYC1 adopt two different rotamer conformations in the 'up' and 'down' states, respectively, and the hypothetical all 'down' subunits would result in widening of the pore that is consistent with an open conformation[25]. Thus, it has been suggested that transition of the 'up' to 'down' configuration occur upon *Dm*MSL10/FLYC1 opening, which is further supported by increased channel activity of a charge reversal mutation that specifically destabilizes the 'up' conformation[25]. This hypothetical gating model of *Dm*MSL10/FLYC1 is consistent with the gating transition observed in our *At*MSL10 structures. First, our

open *At*MSL10 channel forms a symmetric heptamer, and all subunits adopt the 'down' configuration. Second, side chain rearrangement of the equivalent gating phenylalanine (F553) alone in *At*MSL10 gives rise to drastically different opening of the central ion-conduction pore. However, one important difference is noted between these proposed gating mechanisms for *At*MSL10 and *Dm*MSL10/FLYC1. In all our *At*MSL10 structures, the TM4-TM5 linker domain is in the 'down' configuration, likely because of the absence of the corresponding salt bridge that is necessary to stabilize the 'up' conformation in *Dm*MSL10/FLYC1. Thus, this subtle structural difference may fine tune the mechanotransduction process in these two closely related channels.

We have previously elucidated the closed and open structures of a related plant channel, *At*MSL1, by leveraging a gain-of-function mutant and revealed a 'flattening and expansion' gating transition stemming from a highly curved TMD in the resting state[32]. Notably, curved TMDs are also present in the structurally unrelated mammalian mechanosensitive Piezo channels, and an analogous flattening and expansion process seems to underlie Piezo channel gating[48–52]. Flattening of curved TMDs results in an expansion of the in-plane area in the lipid bilayer, which is favored under elevated membrane tension. In these gating transitions, large global structural rearrangements of the TMDs accompany channel opening. In sharp contrast, all the *At*MSL10 structures, including open structures of the wild-type channel in detergents or lipids and a presumably closed conformation of the non-conducting mutant G556V, display a relatively flat transmembrane region similar to that in *Dm*MSL10/FLYC1[25]. Closure of the channel pore could be essentially accomplished by local adjustment of the side chain of a critical residue defining the narrowest pore, while the global conformations of the open and closed states are similar. In the closed state, 'pocket' lipids accommodated by the channel protein are commonly found near the hydrophobic gate of MscS homologs[25,32,41,53], and increased membrane tension promotes departure of the 'pocket' lipids in *Ec*MscS[53]. In analogy, lipid departure from the 'pocket' in *At*MSL10 may lead to side chain rearrangement of the gating phenylalanine to activate the channel. In principle, the drastically different structural features and different degrees of membrane deformation induced by these channels suggest distinct mechanical gating mechanisms.

## Methods

The research complies with all relevant ethical regulations approved by Institutional Biological & Chemical (IBC) Safety Committee at Washington University School of Medicine and at the Icahn School of Medicine at Mount Sinai.

### Cloning, expression, and purification

DNA encoding *Arabidopsis thaliana* MSL10 (*At*MSL10, NCBI: NP_001119212.1) was codon optimized and synthesized (Gene Universal Inc.) and ligated into a pPICZ-B vector, which contains a PreScission protease cleavage site and a GFP-His$_{10}$ tag at the C-terminus. Mutations were generated by site-directed mutagenesis with the primers listed in Supplementary Table 2. The plasmids were transformed into *Pichia pastoris* (strain SMD1163H, Invitrogen) for protein expression for structural studies. For protein expression in *Xenopus* oocytes, the corresponding DNA fragments were ligated into a pGEM vector to generate cRNAs for oocyte injection the mMESSAGE mMACHINE™ T7 Transcription Kit (AM1344, Invitrogen).

For protein production, yeast cells expressing the target proteins were harvested and flash frozen. After milling (Retsch MM400), the cells were resuspended in buffer A (50 mM Tris-HCl pH 8.0 and 150 mM NaCl supplemented with DNase I (D-300-1, GoldBio) and protease inhibitors including 3 µg ml$^{-1}$ aprotinin (A-655-100, GoldBio), 1 mM benzamidine (B-050-100, GoldBio), 100 µg ml$^{-1}$ 4-(2-Aminoethyl) benzenesulfonyl fluoride hydrochloride (A-540-10, GoldBio),

2.5 µg ml$^{-1}$ leupeptin (L-010-100, GoldBio), 1 µg ml$^{-1}$ pepstatin A (P-020-100, GoldBio), and 200 µM phenylmethane sulphonylfluoride (P-470-25, GoldBio). The protein was purified following the membrane preparation protocols[25,31]. The cell mixtures were centrifuged for 10 min at 2500 g, and the resulted supernatant was centrifuged for 1 h at 100,000 g. After discarding the supernatant generated by high-speed centrifugation, the pellet was resuspended and homogenized in buffer A. 1% (w/v) glyco-diosgenin (GDN, GDN101, Anatrace) was applied to the homogenized membrane to extract the protein for 2 h at 4 °C. To remove the cell debris, the mixture was centrifuged for 0.5 h at 30,000 g. The supernatant was mixed with the anti-GFP nanobody bound Glutathione Sepharose® 4B resin (GE Healthcare Life Sciences) for 3 h at 4 °C. The resin was subsequently collected and washed with 10 bed volumes of buffer B (20 mM Tris-HCl pH 8.0, 150 mM NaCl and 85 µM GDN). The GFP tag was removed using PreScission protease overnight at 4 °C. The protein sample was collected, concentrated using a 100 kDa molecular weight cutoff Amicon Ultra concentrator, and further purified on a Superose 6 Increase 10/300 gel filtration column (GE Healthcare Life Sciences) equilibrated with buffer C (20 mM Tris-HCl pH 8.0, 150 mM NaCl and 40 µM GDN). Peak fractions were collected and concentrated to ~7.5 mg ml$^{-1}$ for cryo-EM experiments.

### Saposin lipid nanoparticles reconstitution

Soybean polar lipid extract in chloroform (541602 C, Avanti Polar Lipids, Inc.) was dried under argon and desiccated with vacuum overnight. Buffer containing 20 mM Tris-HCl pH 8.0, 150 mM NaCl and 14 mM n-Dodecyl-β-D-Maltopyranoside (DDM, D310, Anatrace) was added to rehydrate the lipids to 10 mM and sonicated right before use. Saposin was added to the lipids according to a molar ratio of 1:5 and mixed for 1 h at 4 °C. Protein purified from the Glutathione Sepharose® 4B resin was concentrated to ~2 mg ml$^{-1}$ and mixed with saposin-lipid mixture at a molar ratio of 1:20:100. The mixture was rotated for 0.5 h at 4 °C before addition of Bio-Beads SM-2 (1523920, Bio-Rad) at a final volume of ~12.5% (v/v) and rotated overnight at 4 °C. Bio-Beads was removed by centrifugation and the channel reconstituted in saposin lipid nanoparticles was further purified using Superose 6 Increase 10/300 gel filtration column equilibrated with buffer containing 20 mM Tris-HCl pH 8.0 and 150 mM NaCl. Peak fractions were collected and concentrated to ~1.75 mg ml$^{-1}$ for cryo-EM grid preparation.

### Cryo-EM sample preparation and imaging

For the wild-type *At*MSL10 and mutants purified in detergents, 3.5 µl purified protein was applied to the glow discharged Quantifoil R1.2/1.3 Holey Carbon Grids (Q350CR1.3, Electron Microscopy Sciences). After 20 s, the grids were blotted for 2 s at 100% humidity and plunged into liquid ethane using FEI Vitrobot Mark IV (FEI). Grids were loaded into a Glacios Cryo-TEM operating at 200 kV, which is equipped with a Falcon IV detector (ThermoFisher Scientific). Movies were recorded using EPU 2 software (ThermoFisher Scientific) at a magnification of 150k (120k for mutants) and defocus range from −0.8 to −2.4 µm. The wild-type data were collected at a dose rate of 5.42 electrons per Å² per second, and each movie was recorded for 9.77 s with 48 frames. For K539E and G556V, movies were recorded for 7.33 s in 42 frames at a dose rate of 5.98 and 6.07 electrons per Å² per second, respectively.

For *At*MSL10 in saposin lipid nanoparticles, 3.5 µl purified protein was applied to the glow discharged Quantifoil R2/2 Holey Carbon Grids (Q350CR2, Electron Microscopy Sciences). The grids were prepared in the same way as for *At*MSL10 in detergent using FEI Vitrobot Mark IV. Grids were loaded into a Titan Krios Cryo-TEM operating at 300 kV, which is equipped with a Falcon IV detector (ThermoFisher Scientific). Movies were recorded using EPU 2 software at a magnification of 59k and defocus range from −0.8 to −2.4 µm. Data were collected at a dose rate of 4.0 electrons per Å² per second, and each movie was recorded for 13.37 s with 46 frames.

## Image processing and map calculation

Recorded movies were aligned, and dose-weighted using UCSF MotionCor2 1.4.0[54], followed by contrast transfer function (CTF) estimation (GCTF 1.06)[55] in Relion 3.1.2[56]. Low-quality images were removed manually. Reference-free Laplacian-of-Gaussian (LoG) based autopicking[57] was used to pick particles. A total of 648,864 particles for the wild-type *At*MSL10 in detergents and 404,639 particles in saposin lipid nanoparticles, respectively, were extracted and subjected to reference-free 2D classification in Relion3. For K539E, a subset of micrographs was subjected to LoG-based autopicking followed by 2D classification to generate templates for template-based picking. Subsequently, 1,017,121 particles were extracted and subjected to 2D classification. The particles corresponding to the intact channel with different orientations were selected and imported to cryoSPARC 4.2.1[58] to generate a 3D initial model. The initial map was further refined by non-uniform refinement without symmetry imposed and used as the reference for 3D classification in Relion3. The 3D classes containing 163,661 particles for the wild type in detergents, 69,275 particles for the wild type in saposin lipid nanoparticles, and 326,262 particles for K539E, respectively, were selected and imported into cryoSPARC, and subjected to NU-refinement with C1 or C7 symmetry.

For G556V, motion-corrected micrographs were imported into cryoSPARC and subjected to patch-based CTF estimation. Particles were picked using blob-based autopicking and subjected to 2D classification to generate templates for template-based picking. 1,171,913 particles were extracted and subjected to two rounds of 2D classification. Particles corresponding to the intact channel with different orientations were selected to generate the 3D initial model. After heterogeneous refinement, 376,738 particles were selected and subjected to non-uniform refinement with C1 or C7 symmetry. Two rounds of global and local CTF refinement were performed to further improve the resolution.

To further improve the map quality, two half-maps for each reconstruction following non-uniform refinement were imported into deepEMhancer 0.14[59] at highRes mode to generate the maps guiding model building. Local resolution estimates were calculated in cryoSPARC. Masks covering a single protomer in each of the heptameric channels were generated using Chimera 1.14[60] and cryoSPARC. Subsequently, subunit classification was performed in Relion3.

## Model building and coordinate refinement

The *At*MSL10 model predicted by AlphaFold[28] was used as the initial model and adjusted manually in COOT 0.9[61]. The backbone and side chains were refined against the full map using real_space_refine in PHENIX 1.19.2[62]. For building the atomic models of the wild-type channel in saposin lipid nanoparticles and mutants, the structure of the wild type in detergent was used as the initial model and adjusted manually in COOT. The structures were further refined in PHENIX with real_space_refinement. The final atomic models were evaluated using MolProbity[63]. The pore radius analysis was performed using HOLE 2.2.005[36] and the figures were generated with ChimeraX 1.5[64].

## Electrophysiology

*Xenopus laevis* oocytes were purchased from Xenopus1. Oocytes expressing the channels were patched 5–7 days after injection with 70 nl of $1 \mu g \mu l^{-1}$ cRNA stock solutions. All the experiments were carried out using excised inside-out membrane patches in symmetric buffer (60 mM $MgCl_2$, 5 mM HEPES, pH 7.3). To generate lateral tension in the membrane, suction was manually applied to the pipette with an excised patch. Recordings were made with the Axopatch 1D patch-clamp amplifier, the Digidata 1320 digitizer (Molecular Devices), and PM-015R pressure monitor (World Precision Instruments). Data were acquired at 1 kHz, lowpass filtered at 200 or 500 Hz and analyzed with the pClamp software suite 8.2 and 10.7 (Molecular Devices). The pipettes with -2–3 MΩ resistance were fabricated with a Sutter P-96 puller (Sutter Instruments) from the

Kimble Chase soda lime glass. All measurements were carried out at −40 mV membrane potential.

## Reporting summary

Further information on research design is available in the Nature Portfolio Reporting Summary linked to this article.

## Data availability

The cryo-EM maps have been deposited to Electron Microscopy Data Bank with accession codes EMD-41164, EMD-41165, EMD-41166, and EMD-41168. Atomic coordinates have been deposited to the Protein Data Bank (PDB) with accession codes 8TDJ, 8TDK, 8TDL, and 8TDM.

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

## Acknowledgements

This work was partly supported by NIH grant R01 GM143440 (to P.Y.). We thank the staff scientists at Washington University Center for Cellular Imaging for data collection. We thank Dr. Elizabeth Haswell for critical reading of the paper.

## Author contributions

J.Z. performed biochemical preparations, cryo-EM experiments, structure determination and analysis. G.M. conducted electrophysiology experiments. P.Y. conceived and supervised the project. J.Z., G.M., and P.Y. analyzed the results and prepared the paper. Correspondence and requests for materials should be addressed to P.Y.

## Competing interests

The authors declare no competing interests.
