## [Peer Review File · Nature Communications]

Reviewers' Comments:

Reviewer #1:

Remarks to the Author:

In manuscript # NCOMMS-23-34010-T, titled "Open structure and gating of the Arabidopsis mechanosensitive ion channel MSL10", Jingying Zhang et al reported the Cryo-EM structure of AtMSL10, along with the observation that wildtype MSL10 maintained open conformation. A silent mutant likely generates a hydrophobic gate, which may provide a distinct gating mechanism in the MscS family.

The results certainly give an interesting insight into the MscS channel family gating, and therefore it provides a good candidate to publish in Nature Communications. However, despite the authors' attempts to dissect the role of AtMSL10-specific salt bridges in mechanical force sensing and/or mechanotransduction, the mechanism of AtMSL10 of mechanosensitivity is limited. Below are several major or minor comments. We recommend that this manuscript could be published after addressing the major and minor concerns.

Major Comment #1:

As the force sensing mechanism of AtMSL10 is limited, it would be interesting to see whether the gating elements in the EcMscS channel are conserved in the corresponding regions of AtMSL10 by functional or structural study.

Major Comment #2:

The experimental structure determination and data processing methods seem canonical for producing the 3D model. Along with the detailed experimental procedures, the results were well presented along with the necessary sound reasoning behind stated results. However, over 100,000 particles produce a 3.7 Å map of AtMSL10 in detergent, and the side chain density especially that of the TM6 is not so noticeable. We suggest that the author can re-perform the data processing and obtain a better map, at least showing the density of side chains in TM6 more clearly.

Major Comment #3:

We notice the results reveal that G(V)556 may interact with F553, and the author claimed that the F553s form the hydrophobic gate in the closed state. It is better to test the function or structure of the F553 mutants (such as the hydrophilic mutants) to enhance the hydrophobic gate hypothesis.

Minor Comments:

In line 55, please cite the paper "Zhang, M. et al. Structure of the mechanosensitive OSCA channels. *Nat. Struct. Mol. Biol.* 25, 850–858 (2018).", as Zhang, M. et al identified and reported OSCA is the high threshold mechanical force-activated channel before that Elife paper. There are no other particular ethical concerns for the work, and we recommend that this manuscript could be published after addressing the major and minor concerns.

Reviewer #2:

Remarks to the Author:

This manuscript is an interesting paper on the 3D-structure of the mechanosensitive (MS) ion channel MSL10 from *Arabidopsis thaliana*. MSL10 is a fascinating MS ion channel in terms of its structure and function relationship. MSL10 promotes programmed cell death when overexpressed or in response to cell swelling (Veley et al., 2014). This activity is provided by a soluble 164 amino acid residues at the N-terminal region and separatable from its function as an MS ion channel (Maksaev et al., 2018). More interesting from the viewpoint of structural biology, the N-terminal 164 amino acid residues constitute one of intrinsically disordered regions (IDRs) (Flynn et al., 2023), which have now become an attractive topic in plant biology because of their putative roles in various functions, including the stress response. Although the manuscript is considerably well written as a structural biology paper, I believe there is room to enhance the explanation of the background and the results the authors obtained. The followings are point-by-point suggestions.

Line 52: MS ion channels generate not only electrical signals but also ion signals, including calcium signals generated by calcium-permeable MS ion channels.

Line 55: There are at least five well-characterized MS ion channels in plants, such as TPK, OSCA, Piezo, MSL, and MCA. The last one (Nakagawa et al. 2007; Yoshimura et al., 2021) is missing in the manuscript.

Line 74-76: MSL9 is more homologous to MSL10 (75% identity) than Flycatcher1 (48% identity), and MSL9 and MSL10 can form a mixture of homoheptamer and heteroheptamer in vivo (Haswell et al., 2008).

Line 248: Please describe the full name of MD as molecular dynamics.

Fig. 1: The legend to this figure describes "Cryo-EM reconstitution of the wild-type full-length channel in detergent", but it seems to me that the figures 1a, 1b, and 1c do not contain the N-terminal, soluble region composed of 164 amino acid residues (see Extended Data Fig 3). Please clarify this point.

Considering that Nature Communications are read by wide audience, especially plant physiologist in this case, I would recommend the authors to add the explanations into each figure. For example, in Fig. 4, the authors should indicate the region of CTD and TMD in the figures.

It seems to me that the Discussion section is somewhat a repetition of the Results section. I would recommend the authors to explain a physiological relevance of their findings. Addition of the explanation of the IDR of MSL10 would attract wide audience because the IDRs are present in a variety of proteins.

References

Flynn, A. J., Miller, K., Codjoe, J. M., King, M. R., and Haswell, E. S. (2023) Mechcanosensitive ion channls MSL8, MSL9, and MSL10 have environmentally sensitive intrinsically disordered regions with distinct biophysical characteristics in vitro. *Plant Direct* 7:e515

Haswell, E. S., Peyronnet, R., Barbier-Brygoo, H., Meyerowitz, E. M., and Frachisse, J.-M. (2008) Two MscS homologs provide mechanosensitive activities in the Arabidopsis root. *Curr. Biol.* 18, 730-734

Maksaev, G., Shoots, J. M., Ohri, S., and Haswell, E. S. (2018) Nonpolar residues in the presumptive pore-lining helix of mechanosensitive channel MSL10 influence channel behavior and establish a nonconducting function. *Plant Direct* 2, 1-13

Nakagawa, Y., Katagiri, T., Shinozaki, K., Qi, Z., Tatsumi, H., Furuichi, T., Kishigami, A., Sokabe, M., Kojima, I., Sato, S., Kato, T., Tabata, S., Iida, K., Terashima, A., Nakano, M., Ikeda, M., Yamanaka, T., and Iida, H. (2007) Arabidopsis plasma membrane protein crucial for Ca²⁺ influx and touch sensing in roots. *Proc. Natl. Acad. Sci. USA* 104, 3639-3644

Veley, K. M., Maksaev, G., Frick, E. M., January, E., Kloepper, S. C., and Haswell, E. S. (2014). Arabidopsis MSL10 has a regulated cell death signaling activity that is separable from its mechanosensitive ion channel activity. *Plant Cell* 26, 3115-3131

Yoshimura, K., Iida, K., and Iida, H. (2021) MCAs in Arabidopsis are Ca²⁺-permeable mechanosensitive channels inherently sensitive to membrane tension. *Nat. Commun.* 12(1):6074. doi.org/10.1038/s41467-021-26363-z

REVIEWER COMMENTS

Reviewer #1 (Remarks to the Author):

In manuscript # NCOMMS-23-34010-T, titled “Open structure and gating of the Arabidopsis mechanosensitive ion channel MSL10”, Jingying Zhang et al reported the Cryo-EM structure of AtMSL10, along with the observation that wildtype MSL10 maintained open conformation. A silent mutant likely generates a hydrophobic gate, which may provide a distinct gating mechanism in the MscS family.

The results certainly give an interesting insight into the MscS channel family gating, and therefore it provides a good candidate to publish in Nature Communications. However, despite the authors' attempts to dissect the role of AtMSL10-specific salt bridges in mechanical force sensing and/or mechanotransduction, the mechanism of AtMSL10 of mechanosensitivity is limited. Below are several major or minor comments. We recommend that this manuscript could be published after addressing the major and minor concerns.

Point 1: As the force sensing mechanism of AtMSL10 is limited, it would be interesting to see whether the gating elements in the EcMscS channel are conserved in the corresponding regions of AtMSL10 by functional or structural study.

Response: We appreciate this great point raised by the reviewer. Amino acid sequence homology between EcMscS and AtMSL10 channels is surprisingly low and is limited to the pore-lining helix (TM3 in EcMscS and TM6 in AtMSL10) and the immediately followed β -domain. A previous study led by one of our co-authors, Grigory Maksaev (Maksaev et al, *Plant Direct* 2018), has functionally characterized critical regions in TM6 in AtMSL10 (equivalent to TM3 in EcMscS) to compare and contrast their contributions to channel gating. Indeed, pore-lining helix alignment presented at Figure 1A in this paper (also shown in our Extended Data Fig. 3) demonstrates multiple striking differences between EcMscS and AtMSL10, such as poor sequence similarity as well as contrasting residues at equivalent positions. EcMscS contains many small-sized amino acids such as Alanine and Glycine, whereas AtMSL10 contains residues with bulky (F), polar (S, T, Q), and charged (K, E) side chains. The only notable similarities are localized to a short hydrophobic stretch immediately preceding the helical kink (G113 in EcMscS and G556 in AtMSL10). In EcMscS, G113 is crucial for inactivation (Edwards et al., *Biophys J.* 2008) and the G113A mutant does not inactivate under sustained tension (Akitake et al, *Nat Struct Mol Biol* 2007). The equivalent G556A mutation in AtMSL10 essentially rendered a wild-type channel with a slightly lower unitary conductance (Maksaev et al, *Plant Direct* 2018). Moreover, a slightly bulkier mutation G556V in AtMSL10 abolished channel activity (Maksaev et al, *Plant Direct* 2018). Taken together, these observations suggest contrasting gating elements in AtMSL10 and EcMscS.

Reference:

Maksaev G, Shoots JM, Ohri S, Haswell ES. Nonpolar residues in the presumptive pore-lining helix of mechanosensitive channel MSL10 influence channel behavior and establish a nonconducting function. *Plant Direct.* 2018 Jun;2(6):e00059. doi: 10.1002/pld3.59. Epub 2018 Jun 5. PMID: 30506019; PMCID: PMC6261518.

Edwards MD, Bartlett W, Booth IR. Pore mutations of the Escherichia coli MscS channel affect desensitization but not ionic preference. *Biophys J.* 2008 Apr 15;94(8):3003-13. doi: 10.1529/biophysj.107.123448. Epub 2007 Dec 7. PMID: 18065458; PMCID: PMC2275705.

Akitake B, Anishkin A, Liu N, Sukharev S. Straightening and sequential buckling of the pore-lining helices define the gating cycle of MscS. *Nat Struct Mol Biol.* 2007 Dec;14(12):1141-9. doi: 10.1038/nsmb1341. Epub 2007 Nov 25. PMID: 18037888.

Fig. 1a from Maksaev et al, *Plant Direct* 2018

To make this point clear, we have now included a new paragraph on Page 13 in Discussion regarding the contrasting gating behaviors between EcMscS and AtMSL10.

“Amino acid sequence homology between AtMSL10 and the prototypical EcMscS is limited, even in the most conserved regions including the pore-lining helix and subsequent β -domain³⁹. The pore-lining helix TM3 in EcMscS is primarily constituted by hydrophobic residues that are enriched in small-sized amino acids such as alanine and glycine. In contrast, the pore-lining helix TM6 in AtMSL10 contains multiple bulky hydrophobic residues as well as polar or charged residues. The only notable similarities are within a short hydrophobic stretch immediately preceding the helical kink (G113 in EcMscS and G556 in AtMSL10). In EcMscS, G113 is critical for inactivation⁴⁰, and the G113A mutant does not inactivate under application of sustained membrane tension⁴¹. The equivalent G556A mutant in AtMSL10 essentially maintains wild-type channel activity with a slightly lower unitary conductance, and a bulkier mutation G556V abolishes channel activity³⁹. Together these contrasting observations hint at a distinct gating scheme in AtMSL10.”

Point 2: The experimental structure determination and data processing methods seem canonical for producing the 3D model. Along with the detailed experimental procedures, the results were well presented along with the necessary sound reasoning behind stated results. However, over 100,000 particles produce a 3.7 Å map of AtMSL10 in detergent, and the side chain density especially that of the TM6 is not so noticeable. We suggest that the author can re-perform the data processing and obtain a better map, at least showing the density of side chains in TM6 more clearly.

Response: The reviewer raised a good point regarding the cryo-EM densities. In general, the local resolution is much higher in the cytoplasmic domain than that of the transmembrane domain. We have performed extensive data processing strategies (including local refinement, particle polishing, and symmetry expansion etc) but could not further improve the cryo-EM reconstructions in the transmembrane domain. We previously used the unsharpened cryo-EM maps to generate the Extended Data Fig. 2, in which densities are more continuous in the transmembrane domain than the b-factor sharpened maps but show less well-resolved side-chain densities. To improve the side-chain densities for the transmembrane domain, we have now also included b-factor sharpened cryo-EM maps for all the datasets and have also deposited the sharpened maps to Electron Microscopy Data Bank. We have now updated Extended Data Fig. 2 with the sharpened cryo-EM maps to show well-resolved side chain densities for several transmembrane helices including TM4-TM6.

Point 3: We notice the results reveal that G(V)556 may interact with F553, and the author

claimed that the F553s form the hydrophobic gate in the closed state. It is better to test the function or structure of the F553 mutants (such as the hydrophilic mutants) to enhance the hydrophobic gate hypothesis.

Response: We appreciate this good point raised by the reviewer. The critical role of F553 in AtMSL10 gating has been previously characterized by electrophysiology and mutagenesis (Maksaev et al., Plant Direct 2018). For a set of constructs (F553W, WT, F553L, and F553V), it has been demonstrated that the bulkiness of the side chains negatively correlated with unitary conductance ($W \approx F > L$), whereas the F553V mutant abolished channel activity. Therefore, it is also evident that this position is sensitive to mutagenesis as a smaller amino acid, valine, rendered a non-functional channel. The authors concluded that F553 is critical for stability of the open state and controls channel conductance. These results are in line with our findings of F553 functioning as a hydrophobic gate.

We have now included the following in the main text on page 13.

“Substitution of F553 with tryptophan or leucine has demonstrated that the bulkiness of the side chain negatively correlates with unitary conductance, whereas the F553V mutant abolishes channel activity³⁹. Therefore, the F553 position is critically involved in gating and determining unitary conductance, which is in line with our proposition of F553 functioning as a hydrophobic gate.”

Point 4: In line 55, please cite the paper “Zhang, M. et al. Structure of the mechanosensitive OSCA channels. Nat. Struct. Mol. Biol. 25, 850–858 (2018).”, as Zhang, M. et al identified and reported OSCA is the high threshold mechanical force-activated channel before that Elife paper. There are no other particular ethical concerns for the work, and we recommend that this manuscript could be published after addressing the major and minor concerns.

Response: Thank you. This paper is now cited.

Reviewer #2 (Remarks to the Author):

This manuscript is an interesting paper on the 3D-structure of the mechanosensitive (MS) ion channel MSL10 from *Arabidopsis thaliana*. MSL10 is a fascinating MS ion channel in terms of its structure and function relationship. MSL10 promotes programmed cell death when overexpressed or in response to cell swelling (Voley et al., 2014). This activity is provided by a soluble 164 amino acid residues at the N-terminal region and separatable from its function as an MS ion channel (Maksaev et al., 2018). More interesting from the viewpoint of structural biology, the N-terminal 164 amino acid residues constitute one of intrinsically disordered regions (IDRs) (Flynn et al., 2023), which have now become an attractive topic in plant biology because of their putative roles in various functions, including the stress response. Although the manuscript is considerably well written as a structural biology paper, I believe there is room to enhance the explanation of the background and the results the authors obtained. The followings are point-by-point suggestions.

Point 1: Line 52: MS ion channels generate not only electrical signals but also ion signals, including calcium signals generated by calcium-permeable MS ion channels.

Response: We thank the reviewer for this good point. We now states ‘*electrical and chemical signals*’ in line 52.

Point 2: Line 55: There are at least five well-characterized MS ion channels in plants, such as TPK, OSCA, Piezo, MSL, and MCA. The last one (Nakagawa et al. 2007; Yoshimura et al., 2021) is missing in the manuscript.

Response: We thank the reviewer for this point and have now added MCA channels with these references.

Point 3: Line 74-76: MSL9 is more homologous to MSL10 (75% identity) than Flycatcher1 (48% identity), and MSL9 and MSL10 can form a mixture of homoheptamer and heteroheptamer in vivo (Haswell et al., 2008).

Response: We thank the reviewer for this good point. We have revised the main text and now it states the following.

“On the protein sequence level, MSL10 is closely related to MSL9 (75% sequence identity) and forms heteromeric channels with MSL9 in the wild-type plant²⁰. In addition, MSL10 is homologous to Flycatcher1 (DmMSL10/FLYC1, 48% sequence identity)...”

Point 4: Line 248: Please describe the full name of MD as molecular dynamics.

Response: Done. Thanks!

Point 5: Fig. 1: The legend to this figure describes “Cryo-EM reconstitution of the wild-type full-length channel in detergent”, but it seems to me that the figures 1a, 1b, and 1c do not contain the N-terminal, soluble region composed of 164 amino acid residues (see Extended Data Fig 3). Please clarify this point.

Response: The wild-type full-length channel was used in protein purification and cryo-EM structural studies but the N-terminal domain (residues 1-165) was not resolved in the cryo-EM reconstructions. To make it clear, we have now explicitly included the following sentence in the figure legend of Fig. 1.

“Notably, densities for the N-terminal ‘death’ domain (residues 1-165) are not resolved.”

Point 6: Considering that Nature Communications are read by wide audience, especially plant physiologist in this case, I would recommend the authors to add the explanations into each figure. For example, in Fig. 4, the authors should indicate the region of CTD and TMD in the figures.

Response: Done. Thanks!

Point 7: It seems to me that the Discussion section is somewhat a repetition of the Results section. I would recommend the authors to explain a physiological relevance of their findings. Addition of the explanation of the IDR of MSL10 would attract wide audience because the IDRs are present in a variety of proteins.

Response: We thank the reviewer for this great point and have now included the discussion of IDRs on Page 12.

“The full-length AtMSL10 channel was subjected to structural analysis, but the N-terminal ‘death’ domain (NTD), consisting of amino acids 1-165, could not be resolved in the cryo-EM densities. This is consistent with the finding that the NTD is an intrinsically disordered region (IDR) that interacts with the C-terminal domain of MSL10 to regulate cell death signaling and ion channel activity^{43,44}. This notion is also reminiscent of the TRPV4 ion channel that is involved in thermo- and osmoregulation, in which the N-terminal IDR modulates channel activity via dynamic long-range interactions⁴⁵. Together, these observations highlight the critical roles of IDRs in ion channel physiology.”

Reviewers' Comments:

Reviewer #1:

Remarks to the Author:

I think the author has basically answered my concerns and I have no further comments.

Congrats!

Mingfeng